# Depth and temperature preferences of meagre, *Argyrosomus regius*, as revealed by satellite telemetry

**Alexander Claus Winkler** [1,2]*, **Lily Bovim** [1], **Bruno C. L. Macena** [3,4], **Miguel Gandra** [1], **Karim Erzini** [1], **Pedro Afonso** [3,4], **David Abecasis** [1]

**1** Centro de Ciências do Mar (CCMAR), Universidade do Algarve, Faro, Portugal, **2** Department of Ichthyology and Fisheries Science, Rhodes University, Makhanda, South Africa, **3** Institute of Marine Sciences - OKEANOS, University of the Azores, Horta, Portugal, **4** Institute of Marine Research (IMAR), Horta, Portugal

* alexwinkrsa@gmail.com

**Data Availability Statement:** All relevant data are within the manuscript and its Supporting information files.

## Abstract

*Argyrosomus regius* (commonly referred to as meagre), is one of Europe's largest coastal bony fish species and supports important recreational and commercial fisheries in the Atlantic and Mediterranean coasts. Demand for this species, and more recently for their swim bladders, has led to regional population declines and growing importance as an aquaculture species. Despite intense research in captivity, little is known about the spatial ecology of *A. regius*'s wild population, including basic information such as vertical migrations and depth/temperature preferences. Previous research based on indirect data suggests a seasonal habitat shift from shallow to deeper waters, but this has never been validated through direct high-resolution movement data. In this study, we tagged 13 adult *A. regius* with pop-up satellite archival tags in the South of Portugal, which successfully returned data from 11 individuals including high-resolution data from six recovered tags (mean, range: 167 days, 28–301 days). We found that adults of this population spend 95.2% of their time between 5 and 75 m depth (mean ± SD, 30.9m ± 18.3m) and do not venture beyond 125 m. Across seasons, *A. regius* move across water temperatures between 13.3 and 24.8°C with a preferred thermal range between 14 and 18°C where they spent 75.4% of their time. The inferential modelling using this electronic data validated previous hypotheses by showing significant differences between a shallower and warmer summer habitat vs. a deeper and cooler winter habitat. Visual investigation of the diel effects on depth preferences suggests subtle changes in depth use between day and night during the warmer months of the year. We speculate that these patterns are in response to the species' behavioural ecology and physiology, reflecting the seasonal changes in water stratification and presence of prey, as well as on the species reproduction, which results in summer spawning aggregations in shallower areas.

**Funding:** Abecasis D (PI Project applicant):This study was funded by national funds by the Portuguese Foundation for Science and Technology (FCT), through the transitional norm DL57/2016/CP1361/CT0036, and projects UID/Multi/04326/2020, UIDP/04326/2020 and LA/P/0101/2020 and BECORV (PTDC/BIA-BMA/30278/2017). BECORV was also financed by CRESC Algarve 2020 through Portugal 2020 and the European Regional Development Fund (FEDER). Winkler AC(full time researcher on project: national funds by the Portuguese Foundation for Science and Technology (FCT), through the transitional norm DL57/2016/CP1361/CT0036, and projects UID/Multi/04326/2020, UIDP/04326/2020 and LA/P/0101/2020 and BECORV (PTDC/BIA-BMA/30278/2017). BECORV was also financed by CRESC Algarve 2020 through Portugal 2020 and the European Regional Development Fund (FEDER) Bovim L (MsC student): Part of this work was the result of a IMBRSea MSc thesis. The funders had no role in study design, data collection and analysis, decision to publish, or preparation of the manuscript.

**Competing interests:** The authors have declared that no competing interests exist.

## Introduction

Consideration of the vertical habitat of marine organisms is imperative to understanding their ecology in a dynamic three-dimensional environment [1–5]. Marine organisms may encounter greater changes in environmental factors such as temperature, pressure, light, and oxygen moving vertically through the water column than across the same distance horizontally [1]. Because factors such as temperature and depth are well known to shape and constrain foraging, thermoregulation, bioenergetics, and reproduction [1, 6, 7], studying the vertical movement patterns of marine animals can provide valuable insight into their behavioural ecology and physiology.

Within these complexities and environmental interdependencies of the habitat selection, temperature is known to be a strong determinant of fish distribution, meaning that thermal preferences may be one of the main factors to explain and predict a species' response to changes in oceanographic conditions related to weather, season, or climate [8, 9]. As upper ocean temperatures are changing due to climate change [10], understanding a fish's thermal preferences greatly improves our ability to predict the resulting changes in their distribution [8, 9]. Additionally, knowledge of a species' thermal envelope within the water column can improve estimates of catchability and, thus, the population assessments that are typically based in fisheries data [4, 11, 12]. The depth and thermal envelopes occupied by a given species also define their environmental niche and, therefore, the organisms they can predate on, be predated by, or compete with. For example, knowledge of an organism vertical space use is critical to interpret its diet and trophic level from the stable isotope ratios [13].

The charismatic but poorly studied, *Argyrosomus regius* (Asso, 1801), is one of the largest species of the Sciaenidae family, known as drums or croakers [14]. It occurs in the eastern Atlantic from Norway to Congo, the Mediterranean, the western Black Sea and, recently, the Red Sea via invasion through the Suez Canal [15–17]. The species is one of the largest fishes living on the north-eastern Atlantic shelf, reaching over 189 cm total length with anecdotal reports of fish over 100kg [18, 19]. It is an important small scale commercial and recreational fishing resource from the Bay of Biscay to the Gulf of Guinea [19, 20] and is gaining popularity in aquaculture, with production exceeding 23400 tonnes in 2019 [21].

*Argyrosomus regius* inhabit inshore and offshore shelf waters down to 200m and most commonly within 15m to 100 m depth [17]. Adults are thought to live in deeper waters where they presumably feed, while juveniles continue to inhabit estuaries and inshore waters until adulthood [22, 23]. Like many other Sciaenids, adults migrate seasonally into estuaries and shallow coastal waters where they aggregate to spawn [16, 19, 22–24]. These migrations are thought to be triggered by warming sea temperature in the boreal spring, with the spawning period lasting from April or May to the end of July in European waters [18, 19, 24, 25]. However, all previous studies on habitat use of adult wild *A. regius* were based on fisheries dependent data or otolith microchemical analysis which can only provide an indirect indication of migrations and have limited spatial resolution [26]. Furthermore, fisheries dependent data can be influenced by fish catchability or sampling effort [27, 28]. It is, therefore, paramount to use fisheries independent techniques such as the tracking of individual fish to validate and interpret habitat use patterns from fisheries dependent data [27, 28].

To our knowledge no studies have investigated the vertical habitat use patterns of adult *A. regius* however some studies have been conducted on the species' congeners *Argyrosomus coronus* [28] and *Argyrosomus japonicus* [29], using fisheries independent techniques. Studies have also been conducted on similar large, bodied sciaenids, such as *Totoaba macdonaldi* [30] and *Atractoscion nobilis* [31]. The results of these previous studies on closely related species revealed that *A. japonicus* and *A. coronus* partake in seasonal onshore/offshore movements while *T. macdonaldi* and *A. nobilis* display rhythmic vertical diel movements.

The only currently available information on the thermal habitat preferences of meagre is from previous studies conducted in the aquaculture settings which show that *A. regius* are tolerant to wide ranges in temperature (13–28°C) [18, 32, 33], and reaches optimal growth at surprisingly high temperatures (ca. 26°C) [34]. Indeed, the water temperature has been suggested to be the most important factor in determining the timing of reproduction and migration in *A. regius*, suggesting that they are sensitive to changes in water temperature [18, 19, 22].

Understanding the fine scale environmental preferences and limits of *A. regius* may also bear substantial implications for its conservation. Although listed as Least Concern by the International Union for Conservation of Nature (IUCN), *A. regius* has life-history characteristics, many shared with other Sciaenidae, that make them vulnerable to exploitation [35]. More recently, there is a growing demand for dried Sciaenid swim bladders or "maw" as a high valued luxury dried seafood product in China which has pushed certain species to the edge of extinction [36]. This increase in demand and sensitivity to exploitation is promoted by the dependence on coastal and estuarine essential habitats for spawning aggregations, which are both more accessible to fishers and anthropogenically impacted than pelagic waters [16, 19, 22, 37]. Additionally, in the case of *A. regius* these appear to be limited to just six areas within large estuary and deltas: the Gironde (France), Tejo (Portugal), Guadalquivir (Spain), Nile (Egypt), Menderes (Turkey) and Banc D'Arguin (Mauritania) [15]. Such limited and dispersed breeding locations also appears to be the reason behind the unusually high level of genetic fragmentation in the European *A. regius* population, which further adds to their vulnerability [15]. If those areas are the few ones with the right environmental conditions for spawning, and if temperature is indeed critical for the aggregation and spawning onset, then understanding this dependency in the wild could be crucial to identify the threats to the essential spawning habitat of this species. This becomes more critical in the face of climate change, given that these areas, located in shallow habitats and low latitude, are more prone to environmental changes [10].

The aim of this study is to describe the vertical movements and environmental preferences of adult *A. regius* via the use of PSATs deployed over a two-year period off the Southern Portuguese coast. We used both data visualisation techniques and an inferential modelling framework to better understand the seasonal and diel variations in the vertical and thermic envelopes used by the species within the region, and to test previous assumptions on the vertical seasonal migrations of the adults in this poorly studied species.

## Materials and methods

Individual *A. regius* were captured in an anchored tuna trap (armação, or almadraba), a large uncovered pound net off Fuzeta in the southern coast of Portugal (37.017° N, 7.708° W; Fig 1). This trap is set to capture the Atlantic bluefin tuna (*Thunnus thynnus*) migrating to and out of the Mediterranean for spawning but captures a variety of other species, of which *A. regius* is one of the most common [37].

Fish were individually seized by divers within the uplifted holding net, then brought aboard the tagging vessel and placed in a stretcher with running seawater through the gills for oxygenation and their eyes covered with a wet cloth to reduce stress. A small incision was made at the tag insertion site, positioned on the basis of the first dorsal fin (Fig 1). A titanium anchor linked to a pop-up satellite archival tag (MiniPAT, Wildlife Computers, WA, USA) was darted into the dorsal musculature so that it would be anchored between the pterygiophores. The anchor was connected to the tag with coated steel cable and a swivel to decrease drag on the anchor. To reduce the movement of the tag, a second keeper-strap anchor incorporating a loose-fitting steel cable loop around the body of the tag was embedded in the dorsal musculature longitudinally in-line with the first anchor. Total length (TL) was measured before tagging

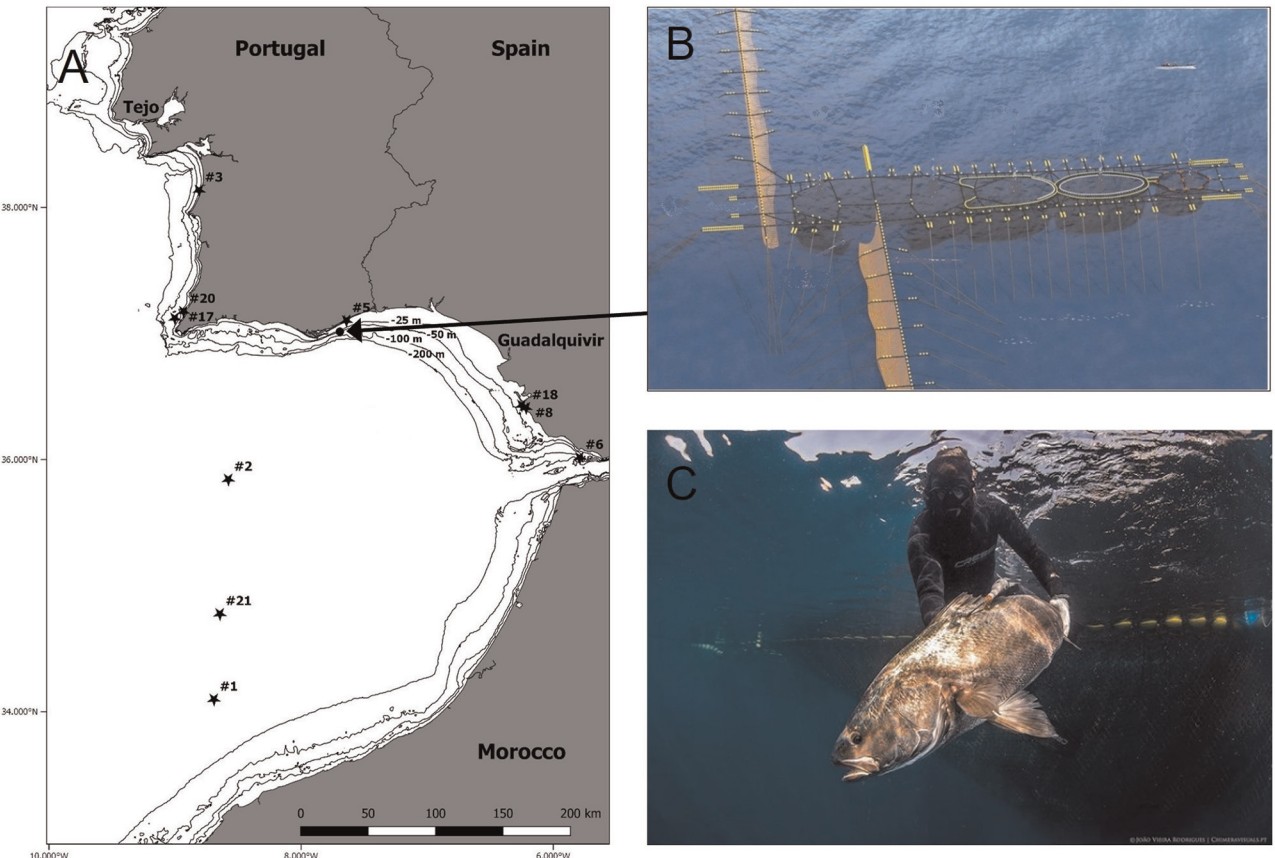

**Fig 1.** A) Map of the study area including coastal bathymetry (black lines). Black circle indicates *Argyrosomus regius* tagging location (tuna trap) and stars denote the pop up/recovery locations of the satellite archival tags with the number indicating the fish ID. Note: Tags from fish #1, #2 and #21 only reported their position weeks after pop-up and thus the positions were obtained after a period of surface drifting. B) The Tunipex tuna trap found off Fuzeta and used to capture adult *A. regius* individuals tagged in this study. C) A diver releasing a PSAT tagged adult *A. regius*.

and only fish above 110 cm TL were selected, meaning that all tagged fish were well above the length of sexual maturity (70–110 cm TL) reported for the species within the Gulf of Cadiz [19] (Table 1). Capture and tagging of *A. regius* was permitted with ICNF (Instituto de Conservação da Natureza e Florestas) permits 560/2018/CAPT and 143/2019/CAPT for the capture, tag, and release of wild fish and DGAV (Direção Geral de Alimentação e Veterinária) permit 0421/000/000/2018 (29/08/2018) for performing experiments with live animals.

The tags were programmed to archive temperature, pressure (depth) and light level every 3 or 5 seconds, depending on the programmed deployment period (Table 1). The MiniPAT depth sensor has a range of 0–1700 m, a resolution of 0.5 m and an accuracy of ±1% of the depth reading. The temperature sensor has a range of -40°C to 60°C, a resolution of 0.05°C and an accuracy of ±0.1°C. Of the six tags deployed in 2018, three had a programmed deployment period of 120 days and the other three of 180 days. The seven tags released in 2019 had programmed deployment periods of 300 days. At the end of this period the pin attaching the tag to the anchor would start to corrode, allowing the tag to disconnect from the anchor, float to the surface and begin to transmit data summaries via Argos satellites constellation. Premature releases may occur if the anchor becomes detached from the fish due to reasons such as poor insertion,

**Table 1. Deployment summary, showing fish ID, total length (TL; in cm), tagging and pop-up dates, number of days attached, mean (± SD) and maximum depth (in m), mean (± SD), minimum and maximum temperature (in ˚C) and data source used in the analysis from the satellite tags deployed on adult *Argyrosomus regius* off the southern coast of Portugal.** * denotes the individuals recaptured. Bold indicate the recovered tags.

| Fish # | TL (cm) | Tagging date | Pop-up date | Programmed days | Days attached | Mean. depth (m) | Max. depth (m) | Mean. Temp (˚C) | Min. temp (˚C) | Max. temp (˚C) |
|---|---|---|---|---|---|---|---|---|---|---|
| 1 | 131 | 2018/09/20 | 2019/01/19 | 120 | 121 | 56.2 ± 16.2 | 112.5 | 17.1 ± 2.1 | 14.4 | 24.2 |
| 2 | 128 | 2018/09/20 | 2019/03/20 | 180 | 181 | 37.6 ± 8.6 | 76.5 | 17.1 ± 2.1 | 14.4 | 23.3 |
| **3** | **131** | **2018/09/20** | **2019/03/05** | **180** | **166** | **36.8 ± 14.4** | **89.5** | **16.2 ± 2.5** | **13.3** | **24.8** |
| **4*** | **132** | **2018/09/20** | **2018/10/18** | **180** | **28** | **39.6 ± 14.5** | **76.0** | **18.5 ± 2.2** | **14.2** | **24.2** |
| **5** | **142** | **2018/09/20** | **2019/01/08** | **120** | **110** | **23.4 ± 8.2** | **64.0** | **17.7 ± 2.2** | **14.5** | **24.8** |
| 6 | 127 | 2018/09/20 | 2019/01/18 | 120 | 120 | 36.9 ± 11.8 | 83.5 | 18.0 ± 2.3 | 14.5 | 24.6 |
| 7 | 126 | 2019/07/09 | No Transmission | 300 | NA | NA | NA | NA | NA | NA |
| **8** | **122** | **2019/07/09** | **2020/05/04** | **300** | **301** | **30.4 ± 21.8** | **125.0** | **16.6 ± 2.3** | **13.3** | **24.8** |
| 17 | 143 | 2019/09/27 | 2020/01/29 | 300 | 124 | 34.2 ± 12.5 | 67.0 | 15.4 ± 1.2 | 13.9 | 19.4 |
| **18** | **131** | **2019/09/27** | **2019/12/19** | **300** | **83** | **34.1 ± 14.7** | **113.0** | **15.9 ± 0.9** | **13.3** | **20.3** |
| 19 | 112 | 2019/09/27 | No Transmission | 300 | NA | NA | NA | NA | NA | NA |
| **20** | **126** | **2019/09/27** | **2020/07/23** | **300** | **300** | **33.5 ± 18.1** | **121.0** | **16.3 ± 2.0** | **13.6** | **23.4** |
| 21 | 135 | 2019/09/27 | 2020/07/23 | 300 | 300 | 46.4 ± 29.9 | 85.5 | 16.5 ± 2.0 | 13.9 | 22.0 |

infection at the anchor insertion site, increased drag as a result of biofouling, entanglement, fish death, etc. or if the tag is removed by a predator or a fisher when the fish is captured [5].

To support the tag's battery lifespan while transmitting data and to optimise data transmission, data summaries may be collected on a periodical basis via "duty cycling". This reduces the number of messages the tag stores so that all the messages can be transmitted in packages before the tag battery expires and when satellites are passing over the tag. The tags deployed in 2019 (all 300 days) were scheduled to store time series (depth and water temperature) messages continuously for the first seven days of deployment and then on alternate days (one day on, one day off) whereas summary profile of depth and temperature (PDT) messages were always generated. The tags deployed in 2018 recorded time series summaries continuously, without a schedule but with summary messages of PDT, time at depth (TAD) and time at temperature (TAT) being generated every six hours. Duty cycling did not affect a tag's full archival dataset, though this is only accessible if the tag (and the archived data) is physically recovered. When tags popped-up and began their Argos transmissions, their locations were tracked and a physical retrieval was attempted using a VHF radio, resulting in six successful recoveries. All the tags were beached when they were recovered.

## Data management and analysis

Where possible, recovered archival data was preferentially used instead of summary time series data (obtained via satellite), considering its higher resolution and coverage. Archival data consisted of data records every 3 to 5 seconds whereas the transmitted summary time series consisted of records every 10 minutes. The first day (24 hours) of data after deployment was excluded to reduce potential post-release effects on the fish behaviour affecting the results. Data were visually examined to confirm the date and time at which the tag detached from the fish (continuous depth readings of 0 m for more than 24 hours) with all data after the beginning of this period being excluded. Where gaps in depth time series data prevented the visual inspection of the surfacing of the tag, the last depth record greater than 2m was chosen as the

final pre-pop-up record. The date and time of this last record was used to calculate the number of days for which the tag was attached to the fish—the difference in days between the tagging and the pop-up dates (or the date of the last relevant data).

Transmitted data were decoded using manufacturer proprietary programs DAP (online manufacturer Portal) and Instrument Helper (for the recovered tags) and analysed with R in R Studio [38]. Depth and temperature time series data from each fish were plotted to examine possible temporal (seasonal) patterns and differences between fish. To improve the visualization of the data a Local Polynomial Regression Fitting (Loess) smoother was added to the plot using loess function [39]. The hist_tad function from R package RchivalTag [40] was used to estimate and plot the proportion of time that *A. regius* spent within 10m depth bins (Time at Depth; TAD) and 1˚C temperature bins (Time at Temperature; TAT). Where time spent is calculated as the number of depth and temperature records within a given bin divided by the total number or records. To facilitate direct comparisons between both archival and transmitted datasets it was decided to exclude all transmitted data days for analysis on which less than 66% of the daily data was received [40].

To statistically assess the effect of season on vertical habitat use, we used the decoded PDTfiles to assess the diving profile throughout the year by selecting the maximum depth bins for each day and fish over all 12 months of year. The Shapiro-Wilk normality test was used to test the normality of the data (W = 0.94872, p-value < 2.2e-16), then a Kruskal-Wallis rank sum test to compare the maximum depths per month, using the package 'stats' [39]. To account for the effect size (i.e. strength of the relationship) we used the interpretation rule of Funder [41] from the package 'effectsize' [42] and, finally, a post-hoc Games-Howell test from the package 'rstatix' [43] to control the Type I errors and increase the power of the analysis when comparing maximum depth differences between each pair of months.

Following the investigation of seasonal depth and temperature patterns, potential diel patterns were visually assessed by averaging temperature and depth readings by month and hour for all archival data, the intersects show the observed values and values in between are smoothed. Contour plots were generated by including a representation of the variation in sunrise, sunset, and crepuscular periods over the studied area. Sunrise, sunset and crepuscular hours were estimated for the study site coordinates, using the 'maptools' package in R [44].

## Results

Tag deployment revealed to be very successful achieving an average of 79% of the programmed deployment period with six tags (46%) remaining attached for the entire programmed deployment. Six tags (Fish #3, #4, #5, #8, #18 and #20) were physically recovered and the raw archival data could be retrieved from them. A further five tags (Fish #1, #2, #6, #17 and #21) successfully popped up and transmitted their data to the Argos satellite constellation and two tags never reported (Fish #7 and #19). This dataset resulted in 26,112,181 datapoints corresponding to a total 1084 days of observations.

In some cases, data packages were not fully transmitted, resulting in gaps in data such as missing days or records of either temperature or depth. Fish #4 was recaptured in the Tunipex tuna trap at Fuzeta (the tagging location) after a month, so this tag did not transmit summary data and only its archival dataset was downloaded. This tag (later re-deployed on Fish #7) and the tag on fish #19 never poppedup and no satellite transmissions were received. Six tags (Fish #1, #2, #6, #8, #20, #21) completed their set deployment period and two (Fish #3 and #5) prematurely released less than 14 days before their programmed release date. Tags #17 and #18 popped up before reaching half of their deployment period. Fish #8 was also recaptured in the net set one year after deployment, (a few months after its tag had popped up) and neither this

fish nor Fish #4 showed evidence of negative impact (e.g., infection) from being tagged. The lengths (TL) of tagged fish that produced data worthy of analysis ranged from 112cm– 143cm with an average size of 131.3 cm (Table 1).

The PSAT tags pop-off locations were restricted to Portugal's south coast,the Gulf of Cádiz and the Strait of Gibraltar, Spain (Fig 1).

All but one (#4) of six fish tagged in 2018 recorded a minimum of 110 days of data for an average 121 days deployment period whereas the five 2019 successfully tagged fish recorded a minimum of 89 data days for an average 222 days deployment period (Table 1, Fig 2). Fall months were covered by all 11 fish except #4, with the remaining months irregularly recorded across individuals: 3 fish (#8, #20, #21) in spring and early summer, and one fish (#8) in summer.

The maximum depths recorded by individual tags ranged between 64m–125m deep (Table 1). The mean depth for all tags combined was 30.9m ± 18.3m ($\bar{\alpha}$ ± SD). On average, fish spent the greatest proportion of their time at depths between 5 m– 74.5 m (95.2%), except #1 which spent 56.4% of its time between 65 m– 84.5.9 m (Table 2). Additionally, all fish spent on average little time shallower than 5 m (0.55%), with one fish (#17) eventually spending 3.18% above this depth. Apart from #1, which displayed an overall deeper distribution than the others, fish spent little time deeper than 64.5 m (Table 2).

Minimum and maximum temperatures encountered by individual tagged fish ranged between 13.3˚C– 14.5˚C and 19.4˚C– 24.8˚C respectively (Table 1). On average fish spent the

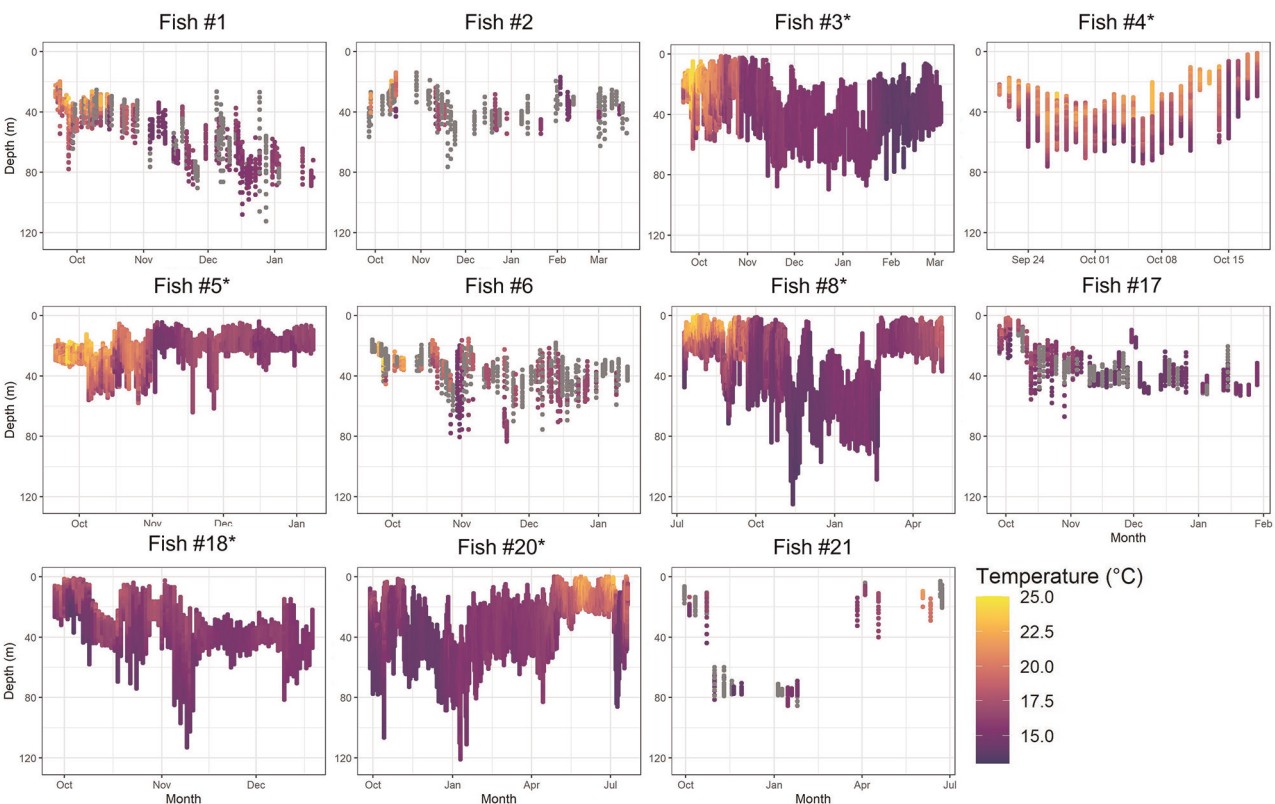

**Fig 2. Temperature and depth profiles for 11 *Argyrosomus regius* tagged with PSAT tags off the Southern Portuguese coast in 2018 (Fish #1,2,3,4,5,6,8) and 2019 (Fish # 17,18,20,21).** Note: grey dots are due to the lack of water temperature data. Tags from fish #3, #4, #5, #8, #18 and #20 were physically recovered and therefore full datasets were obtained instead of transmitted summaries.

**Table 2. Individual fish tagging summary information as well as summarised percentage of time spent within a specific depth and temperature bin.** Data has been conditionally formatted within each column on a scale from green (lowest use) to red (highest use). Data days with fewer than 66% of transmissions have been filtered out, resulting in the exclusion of fish #21. *Denote full archival datasets.

| | Fish ID | Fish #1 | Fish #2 | Fish #3* | Fish #4* | Fish #5* | Fish #6 | Fish #8* | Fish #17 | Fish #18* | Fish #20* | Average |
|---|---|---|---|---|---|---|---|---|---|---|---|---|
| | **Tagging date** | 2018/09/20 | 2018/09/20 | 2018/09/20 | 2018/09/20 | 2018/09/20 | 2018/09/20 | 2019/07/09 | 2019/09/27 | 2019/09/27 | 2019/09/27 | |
| | **Pop-up date** | 2019/01/19 | 2019/03/20 | 2019/03/05 | 2018/10/18 | 2019/01/08 | 2019/01/18 | 2020/05/04 | 2020/01/29 | 2019/12/19 | 2020/07/23 | |
| | **Season** | Aut/Win | Aut/Win/Spr | Aut/Win/Spr | Aut | Aut/Win | Aut/Win | Sum/Aut/Win/Spr | Aut/Win | Aut/Win | Aut/Win/Spr/Sum | |
| | **Days Attached (days)** | 121 | 181 | 166 | 28 | 110 | 120 | 301 | 124 | 83 | 300 | 153.4 |
| | **Fish Size (cm)** | 131 | 128 | 131 | 132 | 142 | 127 | 122 | 143 | 131 | 126 | 131.3 |
| | **No data days** | 24 | 4 | 166 | 28 | 110 | 16 | 301 | 26 | 83 | 298 | 105.6 |
| **Depth Bins (m)** | 0–4.5 | 0.00 | 0.00 | 0.08 | 0.05 | 0.00 | 0.00 | 0.71 | 3.18 | 0.56 | 0.90 | **0.55** |
| | 5–14.5 | 0.00 | 1.04 | 5.03 | 2.94 | 5.06 | 0.00 | 20.33 | 19.41 | 12.37 | 13.99 | **8.02** |
| | 15–24.5 | 0.00 | 44.79 | 14.38 | 12.42 | 43.17 | 4.34 | 19.62 | 26.97 | 23.16 | 14.50 | **20.33** |
| | 25–34.5 | 4.17 | 46.88 | 17.14 | 11.67 | 26.36 | 30.03 | 6.66 | 25.44 | 9.20 | 21.61 | **19.92** |
| | 35–44.5 | 17.53 | 7.29 | 32.58 | 20.36 | 18.67 | 13.02 | 9.43 | 21.27 | 38.27 | 20.41 | **19.88** |
| | 45–54.5 | 9.20 | 0.00 | 22.32 | 39.81 | 6.53 | 20.31 | 15.17 | 3.29 | 9.91 | 13.29 | **13.98** |
| | 55–64.5 | 8.51 | 0.00 | 6.55 | 11.48 | 0.20 | 25.69 | 12.70 | 0.44 | 2.20 | 9.36 | **7.71** |
| | 65–74.5 | 30.38 | 0.00 | 1.83 | 1.28 | 0.00 | 2.95 | 12.79 | 0.00 | 2.41 | 4.54 | **5.62** |
| | 75–84.5 | 26.04 | 0.00 | 0.08 | 0.00 | 0.00 | 3.65 | 1.56 | 0.00 | 0.62 | 0.74 | **3.27** |
| | 85–94.5 | 3.30 | 0.00 | 0.00 | 0.00 | 0.00 | 0.00 | 0.64 | 0.00 | 0.86 | 0.30 | **0.51** |
| | 95–104.5 | 0.69 | 0.00 | 0.00 | 0.00 | 0.00 | 0.00 | 0.33 | 0.00 | 0.42 | 0.26 | **0.17** |
| | 105–114.5 | 0.17 | 0.00 | 0.00 | 0.00 | 0.00 | 0.00 | 0.04 | 0.00 | 0.02 | 0.09 | **0.03** |
| | 115–124.5 | 0.00 | 0.00 | 0.00 | 0.00 | 0.00 | 0.00 | 0.01 | 0.00 | 0.00 | 0.01 | **0.00** |
| | | | | | | | Percentage of time within each depth bin | | | | | |
| **Temperature bins (C°)** | 13–13.99 | 0.00 | 0.00 | 10.44 | 0.00 | 0.00 | 0.00 | 1.60 | 0.13 | 0.51 | 1.57 | **1.43** |
| | 14–14.99 | 15.29 | 26.55 | 16.39 | 3.67 | 3.97 | 13.97 | 23.55 | 47.13 | 13.37 | 24.45 | **18.83** |
| | 15–15.99 | 34.38 | 0.25 | 49.08 | 9.50 | 21.49 | 8.02 | 30.93 | 25.80 | 44.11 | 37.00 | **26.06** |
| | 16–16.99 | 15.66 | 26.05 | 3.81 | 14.46 | 24.14 | 33.87 | 12.45 | 10.73 | 25.69 | 10.81 | **17.77** |
| | 17–17.99 | 8.04 | 28.65 | 1.53 | 15.81 | 18.45 | 14.04 | 9.13 | 11.80 | 14.60 | 5.31 | **12.73** |
| | 18–18.99 | 8.52 | 4.95 | 3.32 | 15.95 | 4.89 | 1.71 | 5.30 | 3.74 | 1.62 | 7.53 | **5.75** |
| | 19–19.99 | 6.69 | 4.18 | 4.27 | 11.90 | 7.47 | 2.57 | 4.62 | 0.67 | 0.10 | 5.99 | **4.84** |
| | 20–20.99 | 4.13 | 5.48 | 3.24 | 12.32 | 9.01 | 4.41 | 4.31 | 0.00 | 0.00 | 3.25 | **4.61** |
| | 21–21.99 | 4.21 | 3.90 | 2.47 | 8.65 | 5.65 | 7.90 | 4.43 | 0.00 | 0.00 | 3.41 | **4.06** |
| | 22–22.99 | 2.68 | 0.00 | 2.50 | 7.52 | 2.49 | 6.01 | 2.85 | 0.00 | 0.00 | 0.66 | **2.47** |
| | 23–23.99 | 0.19 | 0.00 | 2.21 | 0.20 | 1.07 | 2.51 | 0.77 | 0.00 | 0.00 | 0.02 | **0.70** |
| | 24–24.99 | 0.23 | 0.00 | 0.76 | 0.01 | 1.38 | 4.96 | 0.08 | 0.00 | 0.00 | 0.00 | **0.74** |
| | | | | | | | Percentage of time within each temperature bin | | | | | |

greatest proportion of their time at temperatures between 14°C– 18°C (75.4%) and spending more than a quarter of their time (26.1%) between 15–16°C. Fish spent, on average, 23.2% of their time at temperatures above 18ºC and only 1.4% at temperatures below 14º C.

The maximum dive depth per day of tagged *A. regius* significantly differed among the months throughout the year (KS test: chi-squared = 5166.9, df = 8, p-value < 2.2e-16) with an effect of moderate magnitude (Epsilon$^2$ = 0.28, CI 95% = 0.24–1.00). The differences were mainly amongst colder (October to March) and warmer (April to September) seasons (posthoc

**Table 3. P-value results of the Games Howell post-hoc test comparing the number of daily maximum depth dives between months of the year for tagged *Argyrosomus regius*.** Values in parentheses denote the number of daily maximum depth dives used in the analysis, bolded values indicate statistical significance <0.01. Coloured heading correspond to months within each Austral season: spring (green), summer (yellow), autumn (orange), winter (blue).

| Month | Jan (114) | Feb (76) | Mar (55) | Apr (39) | May (12) | Jun (10) | Jul (26) | Aug (31) | Sep (82) | Oct (214) | Nov (183) | Dec (170) |
|---|---|---|---|---|---|---|---|---|---|---|---|---|
| Jan (114) | | 0.831 | **<0.001** | **<0.001** | **<0.001** | **<0.001** | **<0.001** | **<0.001** | **<0.001** | **<0.001** | 0.079 | 0.104 |
| Feb (76) | | | **<0.001** | **<0.001** | **<0.001** | **<0.001** | **<0.001** | **<0.001** | **<0.001** | **<0.001** | 0.999 | 1 |
| Mar (55) | | | | 0.111 | 0.054 | **0.005** | 0.81 | 0.993 | 0.989 | 0.114 | **<0.001** | **<0.001** |
| Apr (39) | | | | | 0.999 | 0.766 | 1 | 0.896 | **<0.001** | **<0.001** | **<0.001** | **<0.001** |
| May (12) | | | | | | 1 | 0.99 | 0.637 | **<0.001** | **<0.001** | **<0.001** | **<0.001** |
| Jun (10) | | | | | | | 0.79 | 0.211 | **<0.001** | **<0.001** | **<0.001** | **<0.001** |
| Jul (26) | | | | | | | | 1 | 0.086 | **<0.001** | **<0.001** | **<0.001** |
| Aug (31) | | | | | | | | | 0.378 | **0.004** | **<0.001** | **<0.001** |
| Sep (82) | | | | | | | | | | 0.368 | **<0.001** | **<0.001** |
| Oct (214) | | | | | | | | | | | **<0.001** | **<0.001** |
| Nov (183) | | | | | | | | | | | | 1 |
| Dec (170) | | | | | | | | | | | | |

test: Game-Howell; Table 3). These findings are further backed up by visual inspection of the data and the corresponding fitted Loess smoother (Fig 3).

Visual inspection of the diel pattern on average depth and temperature revealed subtle diel depth effects during May and June (based on a single individual—#20) and then again during September and October (data from ten fish). Fish utilised the shallowest water during the middle of the day during May and June and utilised deeper water during daylight hours in July, September, and October (Fig 4). These patterns were not expressed by all tagged fish, with fish #20 being the only fish showing a clear utilisation of shallower waters during May and June, as this was the only fish where a significant proportion of its tags deployment time was during May and June (S1 Fig). The utilisation of deeper water during the day in September and October was expressed by fish #3, #8, #18 and #20 (S1 Fig). The diel pattern on temperature utilisation was less well defined with fish generally utilising similar temperature regimes regardless of the time of day.

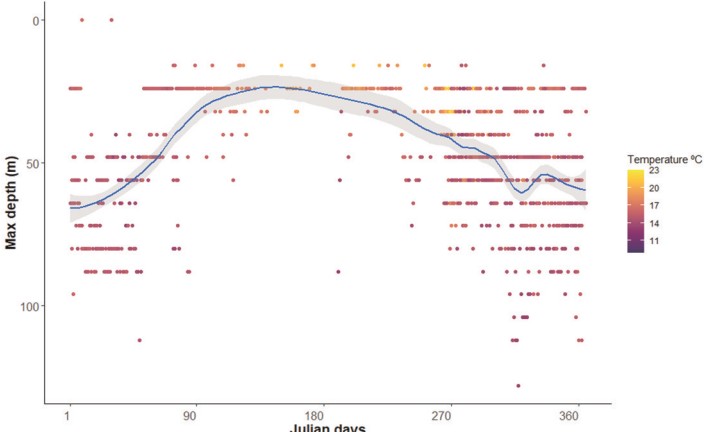

**Fig 3. Daily maximum depth dives and their corresponding temperature for each Julian day covered during this study for tagged *Argyrosomus regius*.** The fitted blue line represents a Loess smoother fitted to data and the shaded area the standard error around the fitted estimate.

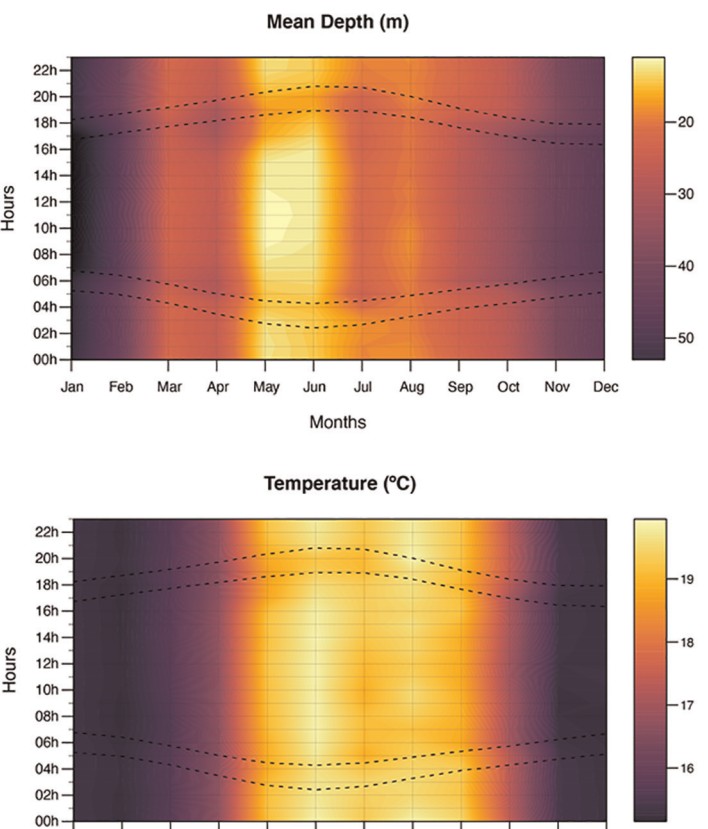

**Fig 4. Contour plots of the average hourly depth and temperatures recorded by six PSAT-tagged *Argyrosomus regius* from which archival data was retrieved for each month of the year.** Dashed lines indicate crepuscular hours around the study site.

## Discussion

This study presents the first fisheries-independent data on high-resolution vertical habitat and temperature preferences of wild *A. regius*. Temperature and depth data were collected via electronic tagging from September 2018 to June 2020, representing all months of the year although weighted towards the autumn and winter months. The main findings suggest those of previous studies which proposed a seasonal shift in vertical habitat use where fish utilise shallower warmer waters during summer and deeper cooler waters during the winter [19]. While not a prominent feature of this dataset certain fish did exhibit marginal diel effects in depth utilisation, utilising shallower depths during the day in May and June and moving deeper during daylight hours in July, September, and October. However, it must be noted that data from May and June is based on a single fish (#20) and from July from two individuals (#8 and #20).

Our archival tagging revealed that *A. regius* indeed inhabits depths within their previously reported preferences (15m to 100m, with 1m to 200m limits) based on fisheries-dependent data [17, 18]. It most frequently inhabited depths of 15 to 44 m and, although they were less frequently present in depths over 65 m, individuals occasionally swam to maximum depths down to 125m. While *A. regius* is described as a demersal species [18, 19], they can be found at the surface and mid-waters and thus changes in depth could be due to the fish changing

location while staying near the seafloor or changing their vertical position in the water column. Regrettably, the precise assessment of demersal or pelagic behaviour using the available dataset presents challenges, demanding the exploration of supplementary methodologies, such as employing fish-attached cameras or acoustic tracking.

Both fishery-dependent data and otolith microchemical assessments suggest that, in the Gulf of Cadiz, *A. regius* migrates offshore to deeper waters in the cooler winter months and into shallower waters around large estuary mouths during summer [19, 22]. The results of our study corroborate this evidence, with fish spending significantly more time at shallow water during the summer and deeper during the winter. Additionally, we found that they dive to deeper depths during the cooler seasons (autumn and winter) when compared to warmer seasons (spring and summer) (See S3 Fig in S1 File). Although our sample is skewed towards the winter months, the two fish (#8, #20) that returned high-quality summer archival data confirmed this trend.

Previous tagging research conducted on two congeners in other regions also found evidence of seasonal changes in vertical habitat use [28, 29]. For example, satellite tagged *A. japonicus* in Australia spend significantly more time deeper (between 20–50m) in the austral autumn and shallower (2–10m) in the summer [29]. Interestingly, acoustically tagged *A. coronus* seems to exhibit the opposite seasonal trend in southern Angola, with animals utilising shallower waters during the austral winter and deeper during summer [28]. A strategy thought to allow *A. coronus* to occupy a preferred narrow thermal range when summer inshore water temperatures exceeded those preferred by this population [28].

Given that *A. coronus* is geographically and phylogenetically *A. regius*'s closest congener, thought to have diverged approximately 1–4 MYA [45], it is interesting that these two species exhibit such different depth, thermal and movement patterns. A pattern that may be a consequence of the unique high latitude environment in which *A. regius* has evolved. While summer shallow warm water habitat use around large estuary mouths such as the Guadalquivir, Tejo and the Gironde in the northeastern Atlantic has been associated with *A. regius* spawning [18, 19, 22], these estuaries are also important summer aggregation and spawning sites for small pelagic fishes. These include the European anchovy *Engraulis encrasicolus* [46] and the sardine *Sardina pilchardus* [47] both small pelagic fishes that can form an important part of adult *A. regius*'s diet [48]. This suggests that *A. regius* may be exhibiting an income breeding strategy, advantageously spawning in areas of high prey abundance so that they have the required resources during gamete production and courtship. Conversely, if adult fish do not feed during courtship, reproductive timing may be more related to egg and larvae survival. For example, *E. encrasicolus* spawning seasonality seems to overlap that of *A. regius* in this area due to increased primary productivity associated with the inflow of the large rivers [23]. This may provide the ideal habitat for larval development, while *A. regius* larvae may be feeding on similar prey items to *E. encrasicolus*, they may also be feeding on *E. encrasicolus* larvae themselves. Larval fish are also known to have a narrow thermal tolerance when compared to adult fish [49] and given that *A. regius* larvae need temperatures above 20–21°C to feed [32] spawning in shallow warm waters may have evolved to aid in larvae survival.

Seasonal changes in regional oceanographic features may also help explain these seasonal differences in vertical habitat and temperature use by *A. regius*. Wind-induced mixing is known to lessen water column stratification in the Gulf of Cadiz during the winter, while water mass stratification intensifies during the summer [50]. This is also evident in individual temperature at depth plots, where fish #3, #5, #8 and #20 which returned high-quality archival data over winter and summer, show stratified depth transects in summer as opposed to winter transects. During the winter, when the waters are less stratified, forages to the surface were rare. Stratification and the presence of warmer inshore surface water within the region during

summer may simply provide a preferred thermal habitat to *A. regius*. Similar findings were found for *Thunnus thynnus* in Mediterranean waters, where fish moved out of shallower waters in the Adriatic Sea when the thermocline broke down at the onset of winter [51]. It must, however, be noted that *A. regius* is a demersal species that is less pelagic in nature when compared to *T. thynnus* and therefore needing to move into shallower coastal waters when the thermocline develops.

There is very little published information on the preferred temperature range of *A. regius*. All available data is derived from aquaculture studies and primarily conducted on larval and juvenile fish [34] that have been acclimated to a variety of controlled temperatures making the results of these studies unreliable for comparative purposes with wild populations, such as in our study [52]. The only information available on adult *A. regius* thermal preferences relates to optimal spawning conditions in captivity [32, 53, 54]. While spawning in captivity is usually induced via a hormonal injection, the best results are obtained when temperatures are maintained between 19–20˚C. Spontaneous spawning has been reported in captivity at temperatures between 20–21˚C during early summer photoperiod regimes. The results from this study do, however, marginally conform with these findings where fish are more likely to inhabit warm (>20˚C) shallow waters (< 30 m) during the spring and summer, which corresponds with the spawning season.

The mechanisms responsible for the utilisation of deeper cooler waters by *A. regius* during winter are, however, less well understood. This is primarily due to the lack of fisheries-dependent information on the species during the boreal winter months when catches are low and adult fish are seemingly utilising deeper areas. A less stratified water column during winter may reduce temperature refugia options available to *A. regius*, explaining the greater heterogeneity in the time spent at specific temperatures and depths during winter months. While Duncan et al. [32] reported that *A. regius* brood stock cease feeding at temperatures below 14˚C, cumulatively, on average tagged fish in this study only spent 1.4% of their time below this temperature. This suggests that *A. regius* is physiologically capable of foraging during the winter despite a lack of information related to their diet during these months. Again, the distribution of one of their primary prey item *E. encrasicolus* [48] may explain their winter vertical distribution. *Engraulis encrasicolus* is known to descend to depths of 100–150 meters during winter corresponding to a vertical habitat shift from warmer surface waters during summer, coinciding with the patterns observed in this study.

The subtle diel vertical movement (DVM) patterns observed for certain fish in this study during the warmer months may also be linked to foraging or spawning. Numerous fish species are known to partake in DVM in response to prey behaviour, usually towards the surface at night and deeper during the day. This has been identified in at least two other sciaenid species; *Totoaba macdonaldi* [30] and *Atractoscion nobilis* [31]. In this study, a similar behaviour was observed but only during the months of September and October for fish #3, #8, #18 and #20, and during July for fish #20, which may suggest that this is related to foraging, as was the case with *T. macdonaldi* [30]. Results regarding the summer months were mostly based on a single fish (# 20). During this period this fish displayed the opposite pattern, moving towards the surface during the midday, particularly during May. Even though this fish's average depth during summer was the lowest of the entire deployment, it did not migrate much deeper at night, amounting to a vertical displacement of <10 meters. Given this subtle effect, it is difficult to hypothesize the cause but since it occurs during the peak spawning season and at the shallowest the fish was found, it may be associated with courtship. This is, however, very speculative and given the low sample size during the summer months, thus should be further investigated in future biotelemetry studies.

While this study has validated the findings of previous studies conducted on this species within the same geographic area [19, 22], there are still numerous questions that need

investigation to fully understand the ecology of this iconic fish species. For instance, very little is known about the feeding habits of adult *A. regius* despite the potential of changes in prey availability to drive the observed changes in seasonal vertical habitat use patterns and some potential evidence of diel foraging behaviour, and thus further studies should evaluate the diet of the species throughout the year. Further effort should also be put into improving the sample size of tagged fish during the entire summer reproductive period. This will help in piecing together why these fish shift their vertical habitat use so drastically between the seasons.

In conclusion, this study found direct evidence that there is a clear seasonal shift in the vertical and thermal envelope of adult *A. regius*. These findings support previous hypotheses based on indirect data such as catch [18, 23] and otolith microchemistry [22]. We also present the first detailed summaries of the preferred temperature and depth ranges of this species in the wild. Summer shallow water utilisation is hypothesised to be in response to the availability of small pelagic fishes, spawning, and a well-defined mixed layer. Rational supporting deeper winter habitat utilisation is less well understood but may be related to reduced water mass stratification and a shift in primary prey distribution to cooler deeper depths. Future studies focusing on the species' dietary habits and the drivers of the interannual movement patterns of individual fish would help us to better understand the species' ecology and, ultimately, management.

## Supporting information

**S1 Fig. Individual contour plots of the average hourly depth and temperatures recorded of six different PSAT tagged *Argyrosomus regius* from which archival data was retrieved for each month of the year.** Dashed lines indicate crepuscular hours around the study site. (PDF)

**S1 File.**
(DOCX)

**S1 Data.**
(PDF)

**S2 Data.**
(PDF)

## Acknowledgments

Tunipex (Alfredo and team), J. Baeyaert, S. Kraft, M. Ramos Martins, and all students that helped during the tagging procedures and tag recovery. Part of this work was the result of an IMBRSea MSc thesis conducted by LB in 2020.

## Author Contributions

**Conceptualization:** Lily Bovim, Pedro Afonso, David Abecasis.

**Data curation:** Bruno C. L. Macena, David Abecasis.

**Formal analysis:** Alexander Claus Winkler, Lily Bovim, Bruno C. L. Macena, Miguel Gandra.

**Funding acquisition:** Karim Erzini, Pedro Afonso, David Abecasis.

**Investigation:** Alexander Claus Winkler, Lily Bovim, David Abecasis.

**Methodology:** Miguel Gandra, Pedro Afonso, David Abecasis.

**Project administration:** Karim Erzini, David Abecasis.

**Resources:** David Abecasis.

**Supervision:** Karim Erzini, Pedro Afonso, David Abecasis.

**Visualization:** Lily Bovim, Bruno C. L. Macena, Miguel Gandra.

**Writing – original draft:** Alexander Claus Winkler, Bruno C. L. Macena, David Abecasis.

**Writing – review & editing:** Alexander Claus Winkler, Karim Erzini, Pedro Afonso, David Abecasis.

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
