## [Decision Letter · Decision Letter 0]

13 May 2022

PONE-D-22-05031Depth and temperature preferences of meagre, Argyrosomus regius, as revealed by satellite telemetryPLOS ONE

Dear Dr. Winkler,

Thank you for submitting your manuscript to PLOS ONE. After careful consideration, we feel that it has merit but does not fully meet PLOS ONE’s publication criteria as it currently stands. Therefore, we invite you to submit a revised version of the manuscript that addresses the points raised during the review process.

We look forward to receiving your revised manuscript.

Kind regards,

Johann Mourier, Ph.D.

Academic Editor

PLOS ONE

Journal Requirements:

Abecasis D (PI Project applicant):This study was funded by national funds by the Portuguese Foundation for Science and Technology (FCT), through the transitional norm DL57/2016/CP1361/CT0036, and projects UID/Multi/04326/2020 and BECORV (PTDC/BIA-BMA/30278/2017). BECORV was also financed by CRESC Algarve 2020 through Portugal 2020 and the European Regional Development Fund (FEDER). 

Winkler AC(full time researcher on project: national funds by the Portuguese Foundation for Science and Technology (FCT), through the transitional norm DL57/2016/CP1361/CT0036, and projects UID/Multi/04326/2020 and BECORV (PTDC/BIA-BMA/30278/2017). BECORV was also financed by CRESC Algarve 2020 through Portugal 2020 and the European Regional Development Fund (FEDER)

Bovim L (MsC student): Part of this work was the result of a IMBRSea MSc thesis

5. We note that Figure 1 in your submission contain map images which may be copyrighted. All PLOS content is published under the Creative Commons Attribution License (CC BY 4.0), which means that the manuscript, images, and Supporting Information files will be freely available online, and any third party is permitted to access, download, copy, distribute, and use these materials in any way, even commercially, with proper attribution. For these reasons, we cannot publish previously copyrighted maps or satellite images created using proprietary data, such as Google software (Google Maps, Street View, and Earth). For more information, see our copyright guidelines: http://journals.plos.org/plosone/s/licenses-and-copyright.

Additional Editor Comments:

Dear Dr. Claus Winkler,

I would like to apologies for the long delay. One reviewer told me that he could send a report but I have not received any news for a few days. As such, in order to avoid any further delay, I decided to serve as a reviewer and take a decision using my opinion and the report of the reviewer who sent a report.

I think the dataset is interesting and provides a real addition to the literature as the behaviour of this species is poorly known. However, I tend to agree with the reviewer that some analyses need some considerations (i.e. GAMM) and that some additional analyses could be also conducted or other details provided (e.g. depth and temperature histograms). Another approach could be the use of Continuous wavelet transformations (CWTs) or FFTs to investigate for cyclical patterns in depth use.

Like the reviewer, I do not understand why depth is found as the response variable (i.e. in depth bin) and in the explanatory variable... this need to be addressed or explained.

Here are additional comments:

Line 56: present only in the Eastern Mediterranean?

Line 208-209: I do not understand the difference between ‘programmed period’ and ‘entire time planned’...

Line 213: check the spaces between the numbers ‘261 121 81’

Figure 1: It could have been nice, if available, to add a picture of these traps and a picture of a tagged meagre (from Figure S1 for example)

As such, I invite you to submit a revised version of your manuscript addressing the reviewer’ comments and additional ones I provided above.

Kind regards

Johann

Reviewers' comments:

Reviewer's Responses to Questions

**Comments to the Author**

1. Is the manuscript technically sound, and do the data support the conclusions?

Reviewer #1: Partly

2. Has the statistical analysis been performed appropriately and rigorously? 

Reviewer #1: No

3. Have the authors made all data underlying the findings in their manuscript fully available?

Reviewer #1: Yes

4. Is the manuscript presented in an intelligible fashion and written in standard English?

Reviewer #1: Yes

5. Review Comments to the Author

Reviewer #1: General comments

This study used satellite telemetry to investigate the seasonal and diel depth and temperature preferences of meagre off the south coast of Portugal. While the data collected is important to understanding the movements and habitat use of this fish and informing its management, I have a number of concerns with the analyses that need to be addressed before I could recommend this manuscript for publication.

Firstly, the statistical analysis exploring the effect of season and water temperature on depth usage is very confusing, and in its current state, seems to be confounded. Having depth as both the response and an explanatory variable is confounded (i.e. not independent). I’m not sure why depth needs to be included as a smoother? Also, given the response is a specific depth bin, shouldn’t there be different smoothers for each specific depth bin?

Secondly, there is more than enough data to take a statistical approach to exploring diel differences in vertical habitat use. Why was this only visually done? Either GAMMs could be used on hourly metrics for recovered tags, or simple t-tests (or non-parametric equivalents) could be conducted.

Thirdly, the way the data is currently visualized is confusing and hard to define the general distribution. Given this is the first report of satellite tag results from this fish species, I think it would be worth having depth and temperature histograms to simply show the mean distributions of the species (and if possible, split into diel sections).

Lastly, given that a relatively high number of tags were recovered, I think it would be worth going into a little more detail about what the fine-scale vertical behaviour of these fish look like. For example, are they oscillating through the water column through the day, or following the contour of the seabed?

More detail is included in the specific comments below.

Specific comments

Abstract

Line 16: This is the only mention of the swim bladders being in demand from this species.. It would be worth disclosing this in the introduction alongside a reason for this demand.

Line 21-23: Add the mean and range deployment duration of the tags here.

Line 23: I do not think you can claim that you found these fish to be ‘strictly coastal’ from the available data as horizontal spatial positions were not investigated. Fish may remain shallow despite being in offshore locations. I understand that this fish is primarily demersal so it is likely it was in shallower coastal locations, but as it cannot be proved from the dataset, so soften these statements.

Introduction

Line 72: Replacing ‘tagging’ with ‘tracking’

Line 88: Replace ‘Redlisted’ with ‘listed’

Materials and Methods:

Line 112: Check consistency of how common names and scientific names are reported throughout. Also check which is used on second mention.

Line 136: Replace ‘deployment period’ with ‘programmed deployment period’

Line 137: as above, replace ‘deployment periods’ with ‘programmed deployment periods’

Line 152: Replace ‘retrieval’ with ‘physical retrieval’

Line 158-159: Specify how data were examined to confirm tag detachment i.e. was this done through investigating the depth record?

Line 166: Delete ‘against time’. Not required as it is already mentioned that these are time series data.

Line 173: How was this cutoff decided on?

Line 175: Was this conducted for all depth bins? And what was the temporal unit of measurement? Daily proportions?

Line 187: Given fish ID has been added as a random effect, this model should technically be referred to as a Generalised Additive Mixed Model (GAMM) at line 175

Line 189: Why is depth being added as an explanatory variable? The response and explanatory variable are therefore not independent… This also goes against the aims of the model stated at line 174 (to evaluate the effect of season and water temperature).

Line 194: What was the cutoff used for this? It’s surprising that temperature and depth were not correlated..

Line 202: Why were these visually assessed? There is more than enough data to evaluate this statistically. For instance, GAMMs could be modelled using hourly vertical metrics, or simple t-tests could be used to explore day versus night distributions for each individual and/or all the data as per Braun et al (2014) and Curnick et al (2020).

Results:

Line 209: Do you have any ideas why early detachment may have occurred? This would be useful for informing future tagging efforts.

Line 230: This is misleading as positions were not estimated (and only deployment and pop-up locations were recorded). It would also be worth noting the class/accuracy of the reported pop-up positions. Also, given that three pop-up positions were recorded offshore (beyond the 200 m depth contour), I think it would be worth softening this statement (I know these three were drifting for a few days, but there is still a possibility that these animals moved offshore).

Lines 243-249: Diel depth histograms would be a great way to visualize and interpret this data.

Lines 243-249: Given access to the archived depth record, it would be great to add some additional detail on the behaviours observed. For instance, are these animals oscillating through the water column at finer temporal scales? Or generally remaining at a level depth while following the seabed?

Discussion:

Line 298: Replace ‘archival’ with ‘satellite’ or ‘electronic’. Some data was transmitted, not archival.

Line 301-302: Soften this slightly. Sample sizes were relatively limited in summer months.

Line 309: delete ‘eventually’. Perhaps replace with ‘occasionally’

Line 310-312: This could be explored and visualized using the archival depth time-series data.

Line 327: As above, need to choose to use scientific or common name on second mention.

Line 369: Delete ‘coastal’ here.

Line 388: watch consistency of capitalization of seasons

Figures and Tables:

Table 3: What is the specific depth bin? And shouldn’t there be unique smoothers for each depth bin modelled?

Histograms of the depth/temperature usage of the tagged individuals would be very useful! This could even perhaps be broken up by season to aid in the interpretation of seasonal differences.

Figure 1: Make the symbol denoting the location of the tuna trap more obvious.

Reference list:

Braun, C.D., Skomal, G.B., Thorrold, S.R., and Berumen, M.L. (2014). Diving behavior of the reef manta ray links coral reefs with adjacent deep pelagic habitats. PLoS ONE 9(2), e88170. doi: 10.1371/journal.pone.0088170.

Curnick, D.J., Andrzejaczek, S., Jacoby, D.M.P., Coffey, D.M., Carlisle, A.B., Chapple , T.K., et al. (2020). Behaviour and ecology of silky sharks around the Chagos Archipelago and evidence of Indian Ocean wide movement. Frontiers in Marine Science. doi: 10.3389/fmars.2020.596619.

6. PLOS authors have the option to publish the peer review history of their article (what does this mean?). If published, this will include your full peer review and any attached files.

Reviewer #1: No

---

## [Author Response · Author response to Decision Letter 0]

28 Jul 2022

Dear Reviewer1,

Thank you for putting a side your time to review this manuscript, your effort, endeavour and attention to detail in our opinion have greatly helped this manuscript. For your ease in reviewing our response I have included your comments and then our responses are in bold following your comment or suggestion. All line numbers referred to in our responses are referencing the document without track changes. Given your request for more figures and analyses I have included extra supplementary information that we feel conforms with our original analyses and have therefore refrained from including them in the main document. If however you feel as though any of this supplementary information should be included in the main document we are happy to include it but feel as though most of it simply backs up our original findings and do not change the papers narrative in any way. 

I hope that you find our responses satisfactory and recommend our revised manuscript for publication in Plosone.

Regards,

Authors,

Reviewer: 

Comments to the Author

1. Is the manuscript technically sound, and do the data support the conclusions?

Reviewer #1: Partly

Reviewer #1: General comments

This study used satellite telemetry to investigate the seasonal and diel depth and temperature preferences of meagre off the south coast of Portugal. While the data collected is important to understanding the movements and habitat use of this fish and informing its management, I have a number of concerns with the analyses that need to be addressed before I could recommend this manuscript for publication.

Authors:

Firstly, the statistical analysis exploring the effect of season and water temperature on depth usage is very confusing, and in its current state, seems to be confounded. Having depth as both the response and an explanatory variable is confounded (i.e. not independent). I’m not sure why depth needs to be included as a smoother? Also, given the response is a specific depth bin, shouldn’t there be different smoothers for each specific depth bin?

Depth was not the response variable, but time spent as a proportion of the deployment by each individual fish was the response variable within each depth bin, depth was then included as a continuous smoother along with all the other explanatory variables. See: https://stat.ethz.ch/pipermail/r-help/2010-March/232580.html.

Reviewer:

Secondly, there is more than enough data to take a statistical approach to exploring diel differences in vertical habitat use. Why was this only visually done? Either GAMMs could be used on hourly metrics for recovered tags, or simple t-tests (or non-parametric equivalents) could be conducted.

Authors:

Unfortunately, the distribution of the data due to overdispersion did not allow us to use the available binomial distributions available in common R packages that run GAMM analyses, this is reason we used a GAM. We decided away from going the T-test route due to the large size of the dataset which can portray significant effects even if miniscule. Please read the following blog regarding the use of such analyses on large data sets where it is stated that: “With a large enough sample size, the hypothesis test can detect an effect that is so minuscule that it is meaningless in a practical sense” (https://statisticsbyjim.com/hypothesis-testing/practical-statistical-significance/). Additionally, please see the included supplementary information Tables S1 & S2 as well as Figures S3 & S4 which summarise the diel data between all fish and how the average diel differences in both depth and temperature are rather meaningless practically even though they are all significant when a nonparametric hypothesis test is conducted. For example the decision to explore the differences in depth distribution on a seasonal basis is made quite clear by the seasonal box plot where magnitudes of monthly differences between depth use on certain individuals was of a magnitude of 20 - 40 meters in certain instances (Fish 1, Fish 3, Fish 8, Fish 17, Fish 20, Fish 21) these are practical meaningful differences that are backed up by the results of the model. The data presented from Fish 21 must however be treated with caution given the lower number of data points when compared to the other fish. We have however presented this fish’s data in the supplementary information given that it is exploratory.

Reviewer:

Thirdly, the way the data is currently visualized is confusing and hard to define the general distribution. Given this is the first report of satellite tag results from this fish species, I think it would be worth having depth and temperature histograms to simply show the mean distributions of the species (and if possible, split into diel sections).

Authors:

We have added both temperature and depth by month and histograms and boxplots split by day and night. We are presenting this data but not doing a statistical analysis as in our opinion the diel differences are practically meaningless given that they range between an average of 1.1 m and 5.7 m with eight fish only showing a diel average mean depth difference of < 2.5 m. Given that these fish utilise depth from 0 to a maxima from 53 – 125 m such small yet significant differences in our opinion are not practically relevant.

Furthermore, in the discussion when we report these subtle differences we say that the fish that showed slight diel differences were very marginal (<10 m) and suggest further study into the behaviour of these fish during the summer spawning season.

This is one way of reporting the data but we do think that the table summarises the data in a way that is easier for a reader to investigate, by easily seeing exactly the percentage of time a fish utilises each temperature and depth bin. Regardless we have added the diel histograms into the appendix

Reviewer:

Lastly, given that a relatively high number of tags were recovered, I think it would be worth going into a little more detail about what the fine-scale vertical behaviour of these fish look like. For example, are they oscillating through the water column through the day, or following the contour of the seabed?

Authors: 

While we are very lucky to have good recovery rate of our tags and could present the finer scale data such as fine scale oscillatory data from a single 24-hour period for each fish in each season, but this will not provide us with any more information to back up our general findings to investigate seasonal depth and temperature utilisation trends. While there are periods of oscillatory swimming behaviour in the dataset there are also periods of level depth swimming. Unfortunately, even if we did report these finer scale results, we would have to make further assumptions on whether these fish are swimming off the bottom or are truly demersal following the seabed’s contours. Fish may swim at a uniform depth off the bottom and not necessarily be hugging benthic contours, oscillatory swimming behaviour may to be fish swimming perpendicular to depth contours while still being close to the bottom swimming up and down benthic features. One way of potentially investigating this would be to know the exact location of the fish and match that up with the best possible bathymetry maps. Again, this would be hard to conduct and would in our opinion not be true, given that light-based geolocation has a very large error margin and based on the results of a modelling process itself. The only way they we would be able to accurately understand whether these fish are swimming midwater or on the bottom would be to attach cameras to fish and observe their direct swimming behaviour. 

Furthermore, there is evidence of oscillatory behaviour in all tagged fish but we have refrained from reporting it as it does not align with our research question on exploring whether archival tagging data aligns with that previously reported from both catch data and otolith stable isotope data. This dataset is incredibly big and this publication would be extensively longer and complicated to follow if these finer scale behavioural components were explored. I totally agree that this would be worth further exploring but we do not feel as though it will add anything to our current conclusions and investigation into the seasonal depth and temperature trends exhibited by this species. We too are of the opinion that investigating the fine-scale vertical use patterns in such a large dataset would be it’s own study. Matching this swimming behaviour to 

Reviewer:

More detail is included in the specific comments below.

Specific comments

Abstract

Line 16: This is the only mention of the swim bladders being in demand from this species.. It would be worth disclosing this in the introduction alongside a reason for this demand.

Authors: 

Lines 91-93: Added to the introduction in prexisting paragraph on the conservation status of the species: ‘More recently, there is a growing demand for dried Sciaenid swim bladders or “maw” as a high valued luxury dried seafood product in China which has pushed certain species to the edge of extinction. ‘

Reviewer:

Line 21-23: Add the mean and range deployment duration of the tags here.

Author:

Line 23- 24: Added: (mean, range: 167 days, 28 – 301 days)

Reviewer:

Line 23: I do not think you can claim that you found these fish to be ‘strictly coastal’ from the available data as horizontal spatial positions were not investigated. Fish may remain shallow despite being in offshore locations. I understand that this fish is primarily demersal so it is likely it was in shallower coastal locations, but as it cannot be proved from the dataset, so soften these statements.

Introduction

Authors:

Line 24 – 25: Changed, you are correct it is impossible to know whether the fish was hugging the bottom or in the upper water levels: ‘We found that adults of this population spend 95.2 % of their time between 5 and 75 m depth (mean ± SD, 30.9m ± 18.3m) and do not venturing beyond 125 m.’

Reviewer:

Line 72: Replacing ‘tagging’ with ‘tracking’

Author:

Line 73: Changed ‘tagging’ to ‘tracking’

Reviewer:

Line 88: Replace ‘Redlisted’ with ‘listed’

Author:

Line 89: changed to listed

Reviewer:

Materials and Methods:

Line 112: Check consistency of how common names and scientific names are reported throughout. Also check which is used on second mention.

Authors:

Changed throughout the paper and only scientific names are used from now on except for the first mention where the common name is reported, please see the tracked changed version for this update

Reviwer:

Line 136: Replace ‘deployment period’ with ‘programmed deployment period’

Author:

Line 137: Changed

Reviwer:

Line 137: as above, replace ‘deployment periods’ with ‘programmed deployment periods’

Author:

Line 140: changed to ‘programmed deployment period’

Reviwer:

Line 152: Replace ‘retrieval’ with ‘physical retrieval’

Author:

Line 156: ‘retrieval’ change to ‘physical recovered’

Reviewer:

Line 158-159: Specify how data were examined to confirm tag detachment i.e. was this done through investigating the depth record?

Author:

Line 163 -165: criteria were added “Data were visually examined to confirm the date and time at which the tag detached from the fish (continuous depth readings of 0 m for more than 24 hours) with all data after the beginning of this period being excluded.”

Reviewer:

Line 166: Delete ‘against time’. Not required as it is already mentioned that these are time series data.

Authors:

Line 171 - 173: changed: ‘Depth and temperature time series data from each fish were plotted to examine possible temporal (seasonal) patterns and differences between fish.’

Reviewer:

Line 173: How was this cutoff decided on?

Authors:

This 66 % criteria was based on conversations with the developer (R Bauer) of the Rchivaltag R package which is generally used as an exclusion criteria amongst PSAT data analysts. While I agree that this criteria is quite arbitrary we did not lack data in this study and therefore conservatively excluded transmitted data given the possibility that incomplete datasets may introduce unintended bias due to uneven data recovery.

Reviewers:

Line 175: Was this conducted for all depth bins? And what was the temporal unit of measurement? Daily proportions?

Authors:

Apologies for not adding enough detail here, given that some of the tags had differing sampling frequencies of 3Hz and 5Hz we calculated the proportion of time spent within each depth bin as the number of daily records divided by the total number of records recorded by the tags.

See lines 181-185: ‘Where time spent was calculated as the number of daily records divided by the total number or deployment records and therefore represented a daily proportion of time spent within a depth bin and corrected for differences in the number of records recorded by tags with different sampling frequencies (3Hz and 5 Hz).’

Reviewer:

Line 187: Given fish ID has been added as a random effect, this model should technically be referred to as a Generalised Additive Mixed Model (GAMM) 

Authors:

Line 175:I must admit that it was my original thinking that you could not include random effects in GAMs but given that the data was so over dispersed and that the data was binomial the package mgcv allows for non-linear modelling and the use of many more response variable data structures than any of the prominent GAMM R packages. I fortunately found some online resources such as this: https://fromthebottomoftheheap.net/2021/02/02/random-effects-in-gams/ which explained the process of including a random effect in a GAM. I therefore decided to go this route even though I do accept that it is quite unconventional, but I feel like it suited the data better than a GAMM which has limited response variable data structures when compared to GAMs in MGCV . More recently there has been more interest into the use of GAMs in the analysis of telemetry data with recent workshop being held by the European Tracking Network (ETN): https://europeantrackingnetwork.org/en/training-school-calculating-positions-aquatic-telemetry

Reviewer:

Line 189: Why is depth being added as an explanatory variable? The response and explanatory variable are therefore not independent… This also goes against the aims of the model stated at line 174 (to evaluate the effect of season and water temperature).

Authors:

Depth bins were changed to continuous variable as depth in ten metered increments corresponding to the time spent at each depth. We were not modelling depth but the time spent at each depth and then depth was added as a explanatory variable as a continuous smoother. 

Line 194: What was the cutoff used for this? It’s surprising that temperature and depth were not correlated..

Authors:

Agreed temperature and depth are usually strongly correlated but given that the largest portion of this data was collected during the boreal winter months when the mixed layer extends to a depth of ~ 125 m there was not much of a relationship between temperature and depth with temperature remaining fairly constant regardless of depth during this period, there was a weaker relationship between temperature and depth during summer but fish remained in the upper water column, VIFs for both depth and temperature smoothers were < 2.5. Given this we decided to include temperature and depth in the same model

Reviewers:

Line 202: Why were these visually assessed? There is more than enough data to evaluate this statistically. For instance, GAMMs could be modelled using hourly vertical metrics, or simple t-tests could be used to explore day versus night distributions for each individual and/or all the data as per Braun et al (2014) and Curnick et al (2020).

Authors: 

As you will see from the tables and box plots the differences in the mean temperature and depth use during the day and night are very small ranging between a 5.7 m – 1.1 difference in mean depth between day and night. Based on the suggestions by the reviewer we too explored the option of using both parametric or non-parametric T-tests to explore whether there was a statistically significant differences between both depth and temperature between day and night. Unsurprisingly the data did not conform with assumptions of a T-test so we conducted Two-sample Kolmogorov-Smirnov (K-S) as suggested by the reviewer, unsurprisingly given the size of the datasets and the very small standard errors significant effects of day and night on depth and temperature use were found for all fish (P< 0.05). We, however, do not think this is an appropriate way of analysing this data given the size of this massive dataset per individual essentially those individuals from which tags were recovered. Given the size of the dataset and based on the size of the standard errors (very small) on most occasion if a diel difference in mean depth corresponds to a difference one or two meters these tests will find a significant effect. The question is whether this is biologically relevant in this study? At the great mean diel differences in depth and temperature of 5.7 m and 0.9 ˚C respectively of a fish that utilises a depth range of 0 – 125 m and fish that are over 1 m in length we do not think that the results of these tests are anything to go by. Please read the answer to the blog regarding the use of such analyses on large data sets where it is stated that: “With a large enough sample size, the hypothesis test can detect an effect that is so minuscule that it is meaningless in a practical sense” (https://statisticsbyjim.com/hypothesis-testing/practical-statistical-significance/)

Based on this we have reported the diel depth effects but refrained from statistically analysing the data and rather subjectively saying that there may be something occurring but we more data from more tagged individual over the summer months when this marginal trend is being observed. This has been highlighted in the discussion and interpretation of our results 

Reviewers:

Results:

Line 209: Do you have any ideas why early detachment may have occurred? This would be useful for informing future tagging efforts.

Authors:

Unfortunately, we do not have any idea of why the detachments occurred, it is impossible to determine this and it can occur due to multiple reasons stipulated in lines 144 – 147.

Reviewer:

Line 230: This is misleading as positions were not estimated (and only deployment and pop-up locations were recorded). It would also be worth noting the class/accuracy of the reported pop-up positions. Also, given that three pop-up positions were recorded offshore (beyond the 200 m depth contour), I think it would be worth softening this statement (I know these three were drifting for a few days, but there is still a possibility that these animals moved offshore).

Author:

Line 238: ‘the continental shelf waters from’ removed to form: ‘The estimated positions of the tagged fish in this study were restricted to Portugal’s central coast (Lisbon area) to the Gulf of Cádiz and the Strait of Gibraltar, Spain (Figure 1). ‘

Information relating to the three tags that popped up offshore Fish 1, 2, 21 is summarised in the below table, for some reason the wet dry sensor on these tags was faulty and these tags only began transmitting their data between two to five weeks after detachment which is logged by the tag. Given this unfortunate event the position of these pop-up location are probably quite far off the actual popup location. Furthermore, the prevailing winds are northerly which would confirm these positions if the tags popped off along the southern Spanish/ Portuguese coast lines. 

Fish Detachment date First transmission date Class Error

Fish 1 18/01/2019 26/02/2019 Class 3 < 250 m

Fish 2 19/03/2019 18/04/2019 Class 1 500 - 1000 m

Fish 21 23/07/2020 07/08/2020 Class 3 < 250 m 

Reviewer:

Lines 243-249: Diel depth histograms would be a great way to visualize and interpret this data.

Authors:

Added as supplementary information see Table S1 & S2, Figure S4 and Figure S6 for diel temperature and depth averages, boxplots and histograms which have been produced and added as supplements.

Reviewer:

Lines 243-249: Given access to the archived depth record, it would be great to add some additional detail on the behaviours observed. For instance, are these animals oscillating through the water column at finer temporal scales? Or generally remaining at a level depth while following the seabed?

Authors:

While we are very lucky to have good recovery rate of our tags and could present the finer scale data such as fine scale oscillatory data from a single 24-hour period for each fish in each season but this will not provide us with any more information to back up our general findings to investigate seasonal depth and temperature utilisation trends. While there are periods of oscillatory swimming behaviour in the dataset there are also periods of level depth swimming. Unfortunately, even if we did report these finer scale results, we would have to make further assumptions on whether these fish are swimming off the bottom or are truly demersal following the seabed’s contours. Fish may swim at a uniform depth off the bottom and not necessarily be hugging benthic contours, oscillatory swimming behaviour may to be fish swimming perpendicular to depth contours while still being close to the bottom swimming up and down benthic features. One way of potentially investigating this would be to know the exact location of the fish and match that up with the best possible bathymetry maps. Again, this would be hard to conduct and would in our opinion not be true, given that light-based geolocation has a very large error margin and based on the results of a modelling process itself. The only way they we would be able to accurately understand whether these fish are swimming midwater or on the bottom would be to attach cameras to fish and observe their direct swimming behaviour. 

Furthermore, there is evidence of oscillatory behaviour in all tagged fish but we have refrained from reporting it as it does not align with our research question on exploring whether archival tagging data aligns with that previously reported from both catch data and otolith stable isotope data. This dataset is incredibly big and this publication would be extensively longer and complicated to follow if these finer scale behavioural components were explored. I totally agree that this would be worth further exploring but we do not feel as though it will add anything to our current conclusions and investigation into the seasonal depth and temperature trends exhibited by this species. We too are of the opinion that investigating the fine-scale vertical use patterns in such a large dataset would be it’s own study. Based on this we have just presented the contour plots to show that there maybe some diel behaviour but it need further investigation with more fish being tagged during summer.

Reviewer:

Discussion:

Line 298: Replace ‘archival’ with ‘satellite’ or ‘electronic’. Some data was transmitted, not archival.

Author:

Line 306: replaced ‘archival’ with ‘electronic’

Reviewer:

Line 301-302: Soften this slightly. Sample sizes were relatively limited in summer months.

Authors:

Line 307 - 309: ‘confirm’ changed to ‘suggested’ and ‘suggested’ changed to ‘proposed’ 

Reviewer:

Line 309: delete ‘eventually’. Perhaps replace with ‘occasionally’

Authors:

Line 316: deleted and changed to ‘occasionally’

Reviewer:

Line 310-312: This could be explored and visualized using the archival depth time-series data.

Authors:

Line 319 – 321: As explained in the response to lines 243-249 above it is still hard from the time series data to confirm this, I have therefore added a sentence explaining the need to incorporate video footage

Reviewers:

Line 327: As above, need to choose to use scientific or common name on second mention.

Authors:

Changed throughout to A. regius 

Reviwer:

Line 369: Delete ‘coastal’ here.

Author:

Line 378: deleted

Reviewer:

Line 388: watch consistency of capitalization of seasons

Author:

Changed throughout from capitalised to lower case

Reviewer:

Figures and Tables:

Table 3: What is the specific depth bin? And shouldn’t there be unique smoothers for each depth bin modelled?

Authors:

Depth bin was coded as continuous as explained above.

Reviewer:

Histograms of the depth/temperature usage of the tagged individuals would be very useful! This could even perhaps be broken up by season to aid in the interpretation of seasonal differences.

Authors:

Added as supplementary information see Figures S6 and S7, Seasonal boxplots have also been added as Figures S2 and S3

Reviewer:

Figure 1: Make the symbol denoting the location of the tuna trap more obvious.

Author:

Arrow added as well as a diagram of the tuna trap and a tagged A. regius

---

## [Decision Letter · Decision Letter 1]

15 Aug 2022

PONE-D-22-05031R1Depth and temperature preferences of meagre, Argyrosomus regius, as revealed by satellite telemetryPLOS ONE

Dear Dr. Winkler,

Thank you for submitting your manuscript to PLOS ONE. After careful consideration, we feel that it has merit but does not fully meet PLOS ONE’s publication criteria as it currently stands. Therefore, we invite you to submit a revised version of the manuscript that addresses the points raised during the review process.

Dear Dr. Winkler,

Thank you for submitting your revised version of “Depth and temperature preferences of meagre, Argyrosomus regius, as revealed by satellite telemetry” to Plos One.

The reviewer found that your manuscript considerably improved from the original version, however, the reviewer remained unconvinced about the inclusion of depth as a smoother in your model.

Could you please address this comment by doing the corrections of the statistical model or clearly explain why the model is correct and why the reviewer may have missed something or misunderstood the model implementation.

Kind regards

Johann

We look forward to receiving your revised manuscript.

Kind regards,

Johann Mourier, Ph.D.

Academic Editor

PLOS ONE

Journal Requirements:

Reviewers' comments:

Reviewer's Responses to Questions

**Comments to the Author**

1. If the authors have adequately addressed your comments raised in a previous round of review and you feel that this manuscript is now acceptable for publication, you may indicate that here to bypass the “Comments to the Author” section, enter your conflict of interest statement in the “Confidential to Editor” section, and submit your "Accept" recommendation.

Reviewer #1: (No Response)

2. Is the manuscript technically sound, and do the data support the conclusions?

Reviewer #1: Yes

3. Has the statistical analysis been performed appropriately and rigorously? 

Reviewer #1: I Don't Know

4. Have the authors made all data underlying the findings in their manuscript fully available?

Reviewer #1: Yes

5. Is the manuscript presented in an intelligible fashion and written in standard English?

Reviewer #1: Yes

6. Review Comments to the Author

Reviewer #1: Thank you for addressing my comments - I can see that the manuscript has substantially improved as a result.

I do, however, remain unconvinced about the inclusion of depth as a smoother in the model. What purpose was this serving? It still seems to me that the response variable is not independant from this explanatory variable (high time in a certain depth bin will inherently be highly correlated with the depth smoother).

7. PLOS authors have the option to publish the peer review history of their article (what does this mean?). If published, this will include your full peer review and any attached files.

Reviewer #1: No

---

## [Author Response · Author response to Decision Letter 1]

28 Mar 2023

Dear reviewer,

In response to your last comment:

” I do, however, remain unconvinced about the inclusion of depth as a smoother in the model. What purpose was this serving? It still seems to me that the response variable is not independant from this explanatory variable (high time in a certain depth bin will inherently be highly correlated with the depth smoother).”

Firstly, we would like to thoroughly apologise for the delay in the resubmission of this manuscript, we acknowledged your concerns regarding the statistical modelling approach used. We agree that the approach initially used had some severe statistical flaws and acknowledge that the response variable and the depth explanatory variable were depended on each other. Unfortunately, given this hurdle we spent a large amount of time debating the right approach. We therefore decided to remove this analysis and use a simpler statistical approach using a Kruskal-Wallis rank sum test after investigating the assumptions equal variance, to compare the maximum depths per month purely based on individual fish maximum diving depth. This was followed by a post-hoc Games-Howell test, to identify statistical significance amongst months. Fortunately, this different approach did not affect the main findings of the manuscript, so we simply replaced the old analysis in the methods and the results with the new simpler analysis. The new figure, post-hoc analysis table and an associated supplementary figure have been added to the manuscript, the previous analysis and associated table have been removed. 

We hope that the omission of the old analysis and the inclusion of the new analysis satisfies your recommendations. We look forward to your response and thoroughly apologise for the delay in the resubmission of the manuscript.

Specific changes made to the manuscript can be found in the track change document.

On behalf of all the authors,

---

## [Decision Letter · Decision Letter 2]

14 Apr 2023

PONE-D-22-05031R2

Depth and temperature preferences of meagre, Argyrosomus regius, as revealed by satellite telemetry

PLOS ONE

Dear Dr. Winkler,

Thank you for submitting your manuscript to PLOS ONE. After careful consideration, we feel that it has merit but does not fully meet PLOS ONE’s publication criteria as it currently stands. Therefore, we invite you to submit a revised version of the manuscript that addresses the points raised during the review process.

Both reviewers feel that this manuscript is a valuable contribution that would deserve publication. One of them, however, has made suggestions that may help improve the final quality of your work, and I would like you to address before final acceptance.

We look forward to receiving your revised manuscript.

Kind regards,

Antonio Medina Guerrero, Ph.D.

Academic Editor

PLOS ONE

Journal Requirements:

Reviewers' comments:

Reviewer's Responses to Questions

**Comments to the Author**

1. If the authors have adequately addressed your comments raised in a previous round of review and you feel that this manuscript is now acceptable for publication, you may indicate that here to bypass the “Comments to the Author” section, enter your conflict of interest statement in the “Confidential to Editor” section, and submit your "Accept" recommendation.

Reviewer #1: All comments have been addressed

Reviewer #2: (No Response)

2. Is the manuscript technically sound, and do the data support the conclusions?

Reviewer #1: (No Response)

Reviewer #2: Yes

3. Has the statistical analysis been performed appropriately and rigorously? 

Reviewer #1: (No Response)

Reviewer #2: Yes

4. Have the authors made all data underlying the findings in their manuscript fully available?

Reviewer #1: (No Response)

Reviewer #2: Yes

5. Is the manuscript presented in an intelligible fashion and written in standard English?

Reviewer #1: (No Response)

Reviewer #2: Yes

6. Review Comments to the Author

Reviewer #1: (No Response)

Reviewer #2: General comments

This study presents a valuable contribution to the knowledge of meagre ecology using satellite tags (though largely based on archived records from physically recovered devices).

This is already a second revision to the original manuscript and it seems the authors have made significant efforts to respond and accommodate the reviewer of the original and R1 submissions. This has been taken into account when carrying out the current review.

Overall, I find the manuscript scientifically sound, once the original reviewer statistical caveats have been addressed. Therefore, I recommend it for publication.

I have some minor comments/suggestions which are indicated below which might help improve some elements of the MS, but most of which are up to the authors to consider (i.e., recommendation for publication is not conditional on those).

Possibly, the major point is that some statements regarding diel patterns could be softened, due to the low sample size or reduced variation in vertical behaviour, notably in summer months.

Additionally, it is a bit strange the authors do not make reference to the horizontal habitat of the fish tracked, noting this information is likely available after the processing through Wildlife Computers data portal. The complex oceanography around the Gulf of Cadiz/Strait of Gibraltar and the different oceanographic regimes in this region, might also help explain some of the vertical patterns observed.

Specific comments

L16 The combination of “demand for this species, and their swim bladders” sounds a bit strange in the abstract without further explanation. Consider removing or rephrasing (e.g., demand for this species, particularly their swim bladders; demand for this species flesh and, in most recent times, swim bladders for the Asian market…).

L25 do not venture

L46 consider: “thermal preferences may be used as one of the main factors to explain” (i.e., it is true temperature explains a lot but there are other factors that condition species distribution- prey abundance, dissolved oxygen..).

L52 Consider the following change: The depth and thermal envelopes occupied by a given species also define their environmental niche… (it is somehow obvious the depth envelope defines the vertical niche, as it is now).

L96 Additionally, in the case of A. regius (the previous sentences seem to be related to sciaenids in general)

L118 The figure caption is embedded in the main text. Figure captions are generally placed elsewhere separately.

L150 Premature releases are generally detected as the tag staying at a constant depth (+- x meters during y days). This can be due to the factors explained, but also to fish death after tagging and it remaining at the seabed. This does not seem to be the case in the current study, but evaluation of fish mortality based on this (or on the breakage of the pin when depth exceeds a maximum depth to prevent implosion) is frequent in large pelagics satellite tagging studies.

L160 It is a very remarkable rate, and most times difficult to achieve in the field. How were tags physically recovered, just verifying they had beached and using Argos transmission location or by means of a radiogoniometer?. Consider specifying a bit the procedure.

L163 Consider changing “quality and quantity” by “resolution and coverage”

L173 Transmitted data

L184 PDT has not been defined before. I imagine it refers to specific files by WC providing depth-temperature profiles, it should be described.

On the other hand, I am not fully aware on how the depths in the PDT file are generated from the archived records (I guess equally spaced from minimum and maximum depths) but if the hours per histogram is not set at 24 h (i.e., if tags are programmed to summarise data in less than 24 h intervals), depths can be biased low for those days when not all histograms are received. Therefore, some further clarification might be beneficial (e.g., if tags were programmed to summarized TAD, TAT and PDT on a daily basis and the way PDT profiles are generated).

L221 (table 1) Fish 8 is not noted as having been recaptured.

L13 The estimated positions refer to the tracks?. If so, some indication on the post-processing (GPE3, WC…) would be desirable. If it refers to tagging and pop-off positions, noting the former is recorded by the taggers and the later is obtained in the argos transmission ,with a relatively high accuracy (e.g., less than 350 in class 2 transmissions), I would remove the word “estimated”.

L229 Consider changing July and August by early and mid-summer, respectively

L245-246 Construction of this sentence seems a bit strange. Consider beginning with: Fish spent on average 23.2% of their time at temperatures above…

L270 The practical totality of what is described in summer months is based on one fish (#20) with just few days from fish #8 in May and July. It is acknowledged in the discussion section, but it should better be also indicated elsewhere, because as it is now it seems to be a conclusion supported by at least several animals.

L272 Shallowest water during the middle of the day… it does not seem very clear from figure 4. I can only observe a slight deeper behaviour around dusk, but it seems mostly constant throughout the day… maybe it is due to the colour palette, but might be worth considering it further (again noting they are data from just one fish).

Moreover, from figure S2, it seems data were averaged by hour and month, not hour and julian day (every tag data begin or end with the month, not in the middle of it, please double-check). Also, sometimes smoothing masks results, hence, in supplementary information, some raw data can be of help also (ie average depth by hour and julian day).

L291 Note data from most May and June originate from one fish, and in July from two.

L299-300 I would just mention demersal or pelagic behaviour would need to be determined using other methodology (I do see other available methods- e.g acoustic tracking) as more suitable for this objective.

L312 Not sure what standar-tagged refers to. Conventional tags?.

L341-343 Does it mean “In winter, in waters with low stratification, forages to the surface were rare”?

L343-344 There are several studies on large pelagic fish (e.g. bluefin tuna) indicating preference to stay over the thermocline particularly at nighttime or when waters are highly stratified.

7. PLOS authors have the option to publish the peer review history of their article (what does this mean?). If published, this will include your full peer review and any attached files.

Reviewer #1: No

Reviewer #2: **Yes: **F.J. Abascal

---

## [Author Response · Author response to Decision Letter 2]

31 May 2023

Dear Reviewer,

We would like to thank you for your comments and suggestions which we feel as though greatly improve the manuscript. We have spent considerable effort in trying to respond to your suggestions which have been specifically outlined below, please refer to the line numbers in the version of the manuscript which lacks track changes. We hope that we have adequality addressed all your comments and hope that the manuscript maybe considered for publication in PlosOne. Our responses to the your comments are highlighted in bold for easier following.

Regards,

Authors,

Specific comments

L16 The combination of “demand for this species, and their swim bladders” sounds a bit strange in the abstract without further explanation. Consider removing or rephrasing (e.g., demand for this species, particularly their swim bladders; demand for this species flesh and, in most recent times, swim bladders for the Asian market…).

See line 16: We have followed the reviewer’s suggestion. This sentence now reads “Demand for this species, and more recently for their swim bladders, has led to regional population declines and growing importance as an aquaculture species”

L25 do not venture 

See line 25: Changed accordingly.

L46 consider: “thermal preferences may be used as one of the main factors to explain” (i.e., it is true temperature explains a lot but there are other factors that condition species distribution- prey abundance, dissolved oxygen..).

See lines 43-46: We rephrased this sentence. It now reads “Within these complexities and environmental interdependencies of the habitat selection, temperature is known to be a strong determinant of fish distribution, meaning that thermal preferences may be one of the main factors to explain and predict a species’ response to changes in oceanographic conditions related to weather, season, or climate [8, 9].”

L52 Consider the following change: The depth and thermal envelopes occupied by a given species also define their environmental niche… (it is somehow obvious the depth envelope defines the vertical niche, as it is now).

See lines 50-52: We agree with this comment and made changes accordingly. The sentence now reads “The depth and thermal envelopes occupied by a given species also define their environmental niche and, therefore, the organisms they can predate on, be predated by, or compete with.”

L96 Additionally, in the case of A. regius (the previous sentences seem to be related to sciaenids in general)

See line 96-98: We agree with this comment and made changes accordingly. The sentence now reads:” Additionally, in the case of A. regius these appear to be limited to just six areas within large estuary and deltas:”

L118 The figure caption is embedded in the main text. Figure captions are generally placed elsewhere separately.

Based on our understanding from the Plosone submission guidelines figure captions are inserted immediately after the paragraph within which it is first cited. The picture below is from the Plosone submission guidelines. 

L150 Premature releases are generally detected as the tag staying at a constant depth (+- x meters during y days). This can be due to the factors explained, but also to fish death after tagging and it remaining at the seabed. This does not seem to be the case in the current study, but evaluation of fish mortality based on this (or on the breakage of the pin when depth exceeds a maximum depth to prevent implosion) is frequent in large pelagics satellite tagging studies.

See lines 147-150: We fully agree with the reviewer. In this sentence we just want to point out what are the most probable causes of premature release. The interpretation of the tags data does not suggest that premature releases were due to fish death, as mentioned by the reviewer. Fish death was added as a possible scenario for early release.

L160 It is a very remarkable rate, and most times difficult to achieve in the field. How were tags physically recovered, just verifying they had beached and using Argos transmission location or by means of a radiogoniometer?. Consider specifying a bit the procedure.

See lines 161-163: We added the following sentence “When tags popped-up and began their Argos transmissions, their locations were tracked and a physical retrieval was attempted using a VHF radio, resulting in six successful recoveries. All tags were beached when they were recovered.”

L163 Consider changing “quality and quantity” by “resolution and coverage” 

See line 166: We followed the reviewer’s suggestion and replaced “quality and quantity” with “resolution and coverage”.

L173 Transmitted data

See line 176:We followed the reviewer’s suggestion and replaced “Transmitter” with “Transmitted”

L184 PDT has not been defined before. I imagine it refers to specific files by WC providing depth-temperature profiles, it should be described.

On the other hand, I am not fully aware on how the depths in the PDT file are generated from the archived records (I guess equally spaced from minimum and maximum depths) but if the hours per histogram is not set at 24 h (i.e., if tags are programmed to summarise data in less than 24 h intervals), depths can be biased low for those days when not all histograms are received. Therefore, some further clarification might be beneficial (e.g., if tags were programmed to summarized TAD, TAT and PDT on a daily basis and the way PDT profiles are generated).

We thank the reviewer for this comment as the changes made have improved the clarity of the methodology and the ms readability.

We made changes to the Materials and Methods (Lines 152-162) to specifically address this comment. 

“To support the tag’s battery lifespan while transmitting data and to optimise data transmission, data summaries may be collected on a periodical basis via “duty cycling”. This reduces the number of messages the tag stores so that all the messages can be transmitted in packages before the tag battery expires and when satellites are passing over the tag. The tags deployed in 2019 (all 300 days) were scheduled to store time series (depth and water temperature) messages continuously for the first seven days of deployment and then on alternate days (one day on, one day off) whereas summary profile of depth and temperature (PDT) messages were always generated. The tags deployed in 2018 recorded time series summaries continuously, without a schedule but with summary messages of PDT, time at depth (TAD) and time at temperature (TAT) being generated every six hours. Duty cycling did not affect a tag’s full archival dataset, though this is only accessible if the tag (and the archived data) is physically recovered.”

L221 (table 1) Fish 8 is not noted as having been recaptured.

Fish 8 was recaptured but well after the study was complete and hence this information was not added to the table as the recapture did not affect the deployment time. the tag was recovered hence bolded and no asterisk.

L223 The estimated positions refer to the tracks?. If so, some indication on the post-processing (GPE3, WC…) would be desirable. If it refers to tagging and pop-off positions, noting the former is recorded by the taggers and the later is obtained in the argos transmission,with a relatively high accuracy (e.g., less than 350 in class 2 transmissions), I would remove the word “estimated”.

See lines 227-228: Figure 1 shows the tagging location and the pop-off locations (based on the argos transmissions). We changed the text to reflect this. It now reads “The PSAT tags pop-off locations were restricted to Portugal’s south coast, the Gulf of Cádiz and the Strait of Gibraltar, Spain (Figure 1).“

L229 Consider changing July and August by early and mid-summer, respectively.

Changed accordingly. See line 233: “in spring and early summer, and one fish (#8) in summer.”

L245-246 Construction of this sentence seems a bit strange. Consider beginning with: Fish spent on average 23.2% of their time at temperatures above…

See lines 249-250: This sentence was replaced with “Fish spent, on average, 23.2% of their time at temperatures above 18ºC and only 1.4% at temperatures below 14º C.”

L270 The practical totality of what is described in summer months is based on one fish (#20) with just few days from fish #8 in May and July. It is acknowledged in the discussion section, but it should better be also indicated elsewhere, because as it is now it seems to be a conclusion supported by at least several animals.

This section was rewritten to better accommodate the issue raised by the reviewer. It now reads in lines 275-276: “effects during May and June (based on a single individual - #20) and then again during September and October (data from ten fish).”

L272 Shallowest water during the middle of the day… it does not seem very clear from figure 4. I can only observe a slight deeper behaviour around dusk, but it seems mostly constant throughout the day… maybe it is due to the colour palette, but might be worth considering it further (again noting they are data from just one fish).

We believe this might be due to the figure resolution. When reproduced in full resolution the lighter yellow between 10 and 12h is well noticed for the months of May and June, when compared to the darker yellow for night-time hours. We added the information that the May and June data is based on a single individual.

Line 275-276: “effects during May and June (based on a single individual - #20) and then again during September and October (data from ten fish)”

Lines 279-280 we added this phase to the end of the sentence: “as this was the only fish where a significant proportion of its tags deployment time was during May and June (Figure S2)”

Moreover, from figure S2, it seems data were averaged by hour and month, not hour and julian day (every tag data begin or end with the month, not in the middle of it, please double-check). Also, sometimes smoothing masks results, hence, in supplementary information, some raw data can be of help also (ie average depth by hour and julian day).

Apologise we have just realised that there were two S2 figures, where in fact the reviewer is referring to figure S1 which is a separate PDF to the other supplementary files. This has been corrected but does not change the interpretation of this figure by the reviewer, we unfortunately labled S1 as S2 in the manuscript submission portal. 

We are however a little confused as the only figure that represents dive depth and julian day is Figure 3. 

We assume the reviewer is referring to Figure S1 as this is the only figure showing individual contour plots between month and hour for each fish (this was previously listed as Figure S2)

Furthermore, the raw data has been plotted in Figure 2

L291 Note data from most May and June originate from one fish, and in July from two.

We added the following sentence in lines 297-298: “However, it must be noted that data from May and June is based on a single fish (#20) and from July from two individuals (#8 and #20).”

L299-300 I would just mention demersal or pelagic behaviour would need to be determined using other methodology (I do see other available methods- e.g acoustic tracking) as more suitable for this objective.

Following the reviewer’s suggestion this sentence now reads in lines 305-307: “Regrettably, the precise assessment of demersal or pelagic behaviour using the available dataset presents challenges, demanding the exploration of supplementary methodologies, such as employing fish-attached cameras or acoustic tracking.”

L312 Not sure what standar-tagged refers to. Conventional tags?.

See line 319: We are referring to acoustic tags. The text was changed accordingly.

L341-343 Does it mean “In winter, in waters with low stratification, forages to the surface were rare”?

Rewritten see lines 348-349: “During the winter, when the waters are less stratrified, forages to the surface were rare.”

L343-344 There are several studies on large pelagic fish (e.g. bluefin tuna) indicating preference to stay over the thermocline particularly at nighttime or when waters are highly stratified.

We thank the reviewer for pointing this out. Our study species is a demersal coastal species, quite different from an oceanic pelagic species such as the bluefin tuna. We have however added two sentences referring to the similarities between these two species when the thermocline breaks down. See lines 351-355: “Similar findings were found for Thunnus thynnus in Mediterranean waters, where fish moved out of shallower waters in the Adriatic Sea when the thermocline broke down at the onset of winter. It must, however, be noted that A. regius is a demersal species that is less pelagic in nature when compared to T. thynnus and therefore needing to move into shallower coastal waters when the thermocline develops.”

---

## [Editor Report · Decision Letter 3]

4 Jul 2023

Depth and temperature preferences of meagre, Argyrosomus regius, as revealed by satellite telemetry

PONE-D-22-05031R3

Dear Dr. Winkler,

We’re pleased to inform you that your manuscript has been judged scientifically suitable for publication and will be formally accepted for publication once it meets all outstanding technical requirements.

Kind regards,

Antonio Medina Guerrero, Ph.D.

Academic Editor

PLOS ONE
---

## [Editor Report · Acceptance letter]

11 Jul 2023

PONE-D-22-05031R3 

Depth and temperature preferences of meagre, Argyrosomus regius, as revealed by satellite telemetry 

Dear Dr. Winkler:

I'm pleased to inform you that your manuscript has been deemed suitable for publication in PLOS ONE. Congratulations! Your manuscript is now with our production department. 

Kind regards, 

on behalf of

Dr. Antonio Medina Guerrero 

Academic Editor

PLOS ONE